# A Circular-Based Reference Point Extraction Method for Correcting the Alignment of Round Parts

**DOI:** 10.3390/s22155859

**Published:** 2022-08-05

**Authors:** Chang Bae Moon, Byeong Man Kim, Dong-Seong Kim

**Affiliations:** 1Department of Smart Electronics, Korea Polytechnic VII (Changwon Campus), Changwon 51518, Gyeongsangnam-do, Korea; 2Computer Software Engineering, Kumoh National Institute of Technology, Gumi 39177, Gyeongbuk, Korea; 3Department of IT Covergence Engineering, Kumoh National Institute of Technology, Gumi 39177, Gyeongbuk, Korea

**Keywords:** round parts, alignment, circular-based reference point extraction method, high-speed

## Abstract

For products such as smartphones, the technology gap between companies is gradually narrowing with the advancements in technology. Therefore, product design can be a visible strategy for differentiation. However, it is difficult to apply automated production and defect detection processes to the various designs that are being developed. This study proposes a high-speed circular measurement method for correcting the alignment of round parts, which is difficult in an automated process. For analyzing the performance of the proposed method, its processing speed and accuracy are compared with those of the existing methods. The results of the analysis indicate that the overall performance of the proposed method is better than those of the existing methods.

## 1. Introduction

For products with high consumer demand, such as smartphones, the technology gap between companies is gradually narrowing with the advances in technology. Hence, product design appears to be a visible strategy for improving a product’s competitiveness and differentiating it from the other products. However, the application of automated production and defect detection to products or parts with various designs is challenging. In the automated production process, rule-based information such as a standardized location or machine vision technology is applied for precise control in assembling the manufactured parts. In rule-based technology, a robot moves to the corresponding position for assembling parts, based on a three-dimensional coordinate input. Machine vision technology analyzes the image of a product or part to enable a robot to assemble the parts or to determine whether a product is defective.

Machine vision technologies can be classified into two categories: those applied before and after production. The former are applied before production for assembling the parts, whereas the latter are applied after production for detecting whether a product is defective. One of the essential techniques in the two categories is alignment correction. In general, this is a technique for judging the angle and position using an engraved mark or unique pattern of the product. However, with parts that have a round design or glass parts, such as in smartphones, there are cases where unique patterns or marks cannot be engraved due to the structural characteristics.

There are two important factors in machine vision technology: the accuracy and processing speed. If the accuracy is not ensured, a product with defective parts may be classified as defective as a whole; if the processing speed is not ensured, production may decrease. Thus, accuracy and processing speed are essential in machine vision technology. In view of the above, this study proposes a prototype-based reference point extraction method to enhance the accuracy and processing speed for the purpose of aligning round parts.

The feature extraction methods for alignment correction involve the use of a line [1,2], integral histograms [3,4], projection-based integral histograms [5], the long axis [6], Hough Line-based reference feature extraction [7], Harris Corner-based feature extraction [8], and Moravec Corner-based feature extraction [9]. The method involving the use of a line, which is a feature included in the part, has the advantage of rapidly correcting the alignment of products with line components; however, it is difficult to apply to round-shaped parts. The integral histogram method detects the marks on products through template matching; however, it is difficult to apply a mark to certain products such as the glass of a smartphone. The long-axis method is applied when the morpheme of the product is atypical, but the processing time is considerable even for products with a simple shape, such as the round parts of smartphones. The Hough Line [7], Harris Corner [8], and Moravec Corner [9] methods also have the disadvantage of long processing times.

In view of the above, this study proposes a circular-based feature extraction method that can be applied both before and after production in machine vision technology for the correcting the alignment of products such as the round parts of smartphones. The remainder of this article is structured as follows. Chapter 2 indicates the systems to which the proposed method can be applied, and Chapter 3 describes the proposed circle-based feature extraction method. Chapter 4 compares the accuracy and processing speed of the proposed method with those of the existing methods, and finally, Chapter 5 concludes the study.

## 2. Structure of the Assembly and Inspection Process System

The method proposed in this study can be applied to two processes: the process of assembling products in the assembly area when the parts are inputted as shown on the left of Figure 1, and the process of judging the presence or absence of defects in the inspection area when products are inputted as shown on the right of Figure 1. This inspection process determines the presence of defects related to the appearance or assembly state and not the function of the product [10,11]. The method proposed in this study performs alignment correction for round-shaped products as well, as shown in Figure 2. It is considered to be a markerless method because it does not require a unique pattern or marker on the product.

The assembly and inspection process system is mainly composed of an inspection device or image processing module, a PLC module, a lighting module, and a camera module. The functions of each module are as follows:Lighting module: This module adjusts the amount of light. When a product is placed in the shooting area, the module provides the required level of light for shooting and is controlled by the inspector/image processing module.Camera module: When a product is placed in the classifier or image processor inspection area, this module captures the product and sends the image to the classifier or image processor module.Classifier or image processing module: This module receives an image for reading from the camera module and sends the control information to the PLC after calculating the quality of the product or the alignment correction value using the received image.PLC module: This module receives the image reception and image shooting information from the classifier or image processor module and lighting module, and controls the motor on the basis of the information received from the classifier or image processor module.

In the detailed system structure related to the alignment correction of products shown on the left of Figure 3, the angle and movement position for alignment correction are calculated using the first image input, and the calculated angle is used for controlling the motor. In the detailed system structure related to the inspection process shown on the right of Figure 3, the ROI is extracted for the first image inputted after alignment correction, and a machine learning process such as a CNN is applied to determine whether the product is defective. The process commonly applied to both systems is shown in the center of Figure 3, where the method proposed in this study corresponds to Step 1.1.1.

## 3. Circle-Based Reference Point Extraction Method for the Correcting the Alignment of Round Parts

Reference point extraction involves two processes: preprocessing and searching for a reference point, as depicted in Figure 4. When the first image is used as input, to find the reference point, preprocessing is performed in the following order: setting the ROI area, searching for the starting point to search for the reference point, and generating a circular coordinate array from the starting point. If the results of checking indicate that the current position is the reference point in the reference point search process, the reference point (the central coordinates of the circle) is returned. If the current position is not the reference point, the pixel is moved (left or right along the axis) to update the circular coordinate array, followed by the process of rechecking whether the position moved to is the reference point.

### 3.1. Concept of the Proposed Algorithm

The proposed algorithm extracts circle-based reference points for correcting the alignment of object designs comprising a mixture of curved components and line components as shown in Figure 5a,b, as well as object designs comprising curved components as shown in Figure 5c. Among the existing methods, line-based reference point extraction can extract the reference point using the line components in the object (Figure 5b); however, it has difficulties in extracting the reference point for an object composed of curved components (Figure 5c). In view of the above, this study proposes a circle-based reference point extraction method that can be applied to both type of objects (Figure 5b,c).

The proposed template for extracting the reference point has a circular shape, as depicted in Figure 6a, consisting of the top, bottom, left, and right areas, and a center point (Position A, i.e., the center of the circle). Position B in the top area or Position C in the bottom area is moved (to the left or right) according to the location of the search starting point and is used for determining the reference point depending on whether the reference point is to the left or right. For example, in Figure 6b, as the starting point is at the bottom, the point corresponding to Template C moves along the boundary line (Path A in Figure 6b) from the start (the start of the search in Figure 6b) to the end (the end of search in Figure 6b), and the algorithm checks whether the reference point exists. As the reference point is to the right of the starting point in Figure 6b, the correct reference point is determined using the coordinates in the right-hand area of the template. For each movement, the coordinate value corresponding to Position C becomes the boundary line’s coordinate value, and the coordinate values of the right-hand area also change accordingly.

### 3.2. Preprocessing

The preprocessing process of the proposed method includes the following steps: ROI area setting, starting point search, and circular coordinate array generation, as shown in Figure 7. Finally, the circular coordinate array is constructed.

(1)ROI Area Setting

In the ROI area setting step, a standard image is provided to the user, as shown in Figure 8a, in which the ROI area is set. In the figure, the red box indicates the ROI area. The coordinate (x, y) is the starting point of the ROI area, and w and h are the size of the ROI area.

(2)Starting Point Search

The starting point search requires a user-inputted point for commencing the search. A and B are examples of the user-inputted points in Figure 8b. When the search is based on Point A, the position where the pixel value $p_{x,\;y}\neq p_{x,\;y+1}\;$increases by 1 from the position of Point A along the *y*-axis is defined as the starting point. When the search is based on Point B, the point where it decreases by unity from the position of Point B along the y-axis and the position with the pixel value $p_{x,\;y}\neq p_{x,\;y-1}\;$is defined as the starting point. The center of the circle (cx, cy) is calculated as (x, y+r) by adding the radius r to the starting point (x, y).

(3)Circular Coordinate Array Generation

The circular coordinate array comprises four circular coordinate arrays (corresponding to the top, bottom, left, and right), as shown on the right of Figure 9. The first half of each array consists of the coordinates of the circular path points corresponding to 45°, obtained using the Midpoint Circle Algorithm [12], and the second half consists of coordinates corresponding to −45°. Thus, the value of Position A in the circular coordinate array corresponding to the top area on the right of Figure 9 indicates the coordinates of A on the left. The value of Position B indicates the coordinates of B on the left, and similarly for Positions C, D, E, F, G, and H in the right-hand circular coordinate array. Furthermore, the circular coordinate arrays of the top area (T), the bottom area (B), the right-hand area (R), and the left-hand area (L) have coordinates in the range of −45°<T<45° with respect to the top at 0°, 135°<B<225°, 45°<R<135°, and 225°<L<315°, respectively.

The pseudocode for generating a circular coordinate array is shown in Figure 10. As the input parameters of the Midpoint Circle Array Generation Method (MCAGM), the central coordinate of a circle, the radius of the circle, and the circular coordinate to be generated are given. The given circular coordinate array is constructed using the center coordinates of the circle (cx, cy) and the difference coordinates (x, y) from the center. The algorithm below is used to change the part that draws the circle in Bresenham’s circle drawing algorithm [13] to the part that stores the coordinates of the circle. That is, Lines 1.3 and 1.8 are changed and the function set_Circle_Cooedinate() is added. Line 2.1 in Figure 10 defines the upper coordinate array of the circle (top in Figure 9). Lines 2.2, 2.3, and 2.4 define the bottom coordinate array (bottom in Figure 9), the left coordinate array (left in Figure 9), and the right coordinate array of the circle (right in Figure 9), respectively.

### 3.3. Reference Point Search

As shown in Figure 11, the reference point search process proceeds by checking the right-hand (or left-hand) side of the circular coordinate array and determining whether the reference point search has been successful. If the search is successful, it terminates after returning the center coordinates. If the search fails, the circular coordinate array is updated after shifting the pixel to the right (or left), followed by the process of checking the right-hand (or left-hand) side of the circular coordinate array.

The pseudocode for searching for the lower-right reference point using the circular coordinate array is shown in Figure 12. The circle center coordinate (cx, cy) and the coordinate array carray [i, len] are the inputs. In Figure 12, the check_Right_Side function (Line 3) determines whether the center coordinate of the location is the reference point. If it is the reference point, the search for the reference point is terminated by returning the center coordinate (cx, cy). If not, the gap between the next position moves and the current position is calculated by the move_Bottom_Right_Position function (Line 4) to continue searching for the reference point. Moreover, the gap value is reflected in the circular coordinate array through the update_Carray function (Line 5) and the center coordinate (cx, cy) through Line 6.

(i)Checking the right-hand (or left-hand) side of the circular coordinate array and determining whether the search for the reference point has been successful

The method for searching for the reference point of the lower-right area based on the circular coordinate array is as shown in Figure 13a, where a point (x, y) in the circular coordinate array is checked regarding whether its pixel value is not same as that of the position to its right (x+1, y). If such a point exists, the search is successful. Figure 13b depicts a case where the search is successful, whereas Figure 13c shows a case where a continued search is required. If the search is successful, the search terminates by returning the coordinate$\;\left(x^{C},\;y^{C}\right)$, which is the center C of the circle. Thus, the center coordinate $\left(x^{C},\;y^{C}\right)$ is the reference point.

The pseudocode for searching the lower-right area based on the circular coordinate array is shown in Figure 14, where the coordinate array carray [i, len] is used as the input parameter. The values x and y are set to one of coordinates of the given circular coordinate array, as in Line 3, and the pixel value of the coordinate (x, y) and that of the coordinate (x+1, y) are checked to ensure that they are different. That is, it is checked to ensure it satisfies$\;\mathrm{image}\_{\mathrm{array}}_{\left(x,\;y\right)}\neq \mathrm{image}\_{\mathrm{array}}_{\left(x+1,\;y\right)}$. The position satisfying this condition is defined as the reference point (see Line 4 and Figure 13b). If the condition is not satisfied, checking of the next coordinate is performed until end of the given circular coordinate array.

(ii)Moving the pixel coordinates to the right (or left) and updating the circular coordinate array

If the right-hand area search fails, the search continues by moving right (or left) from the current position. First, it moves from the current position (A in Figure 15a) to the right (or left) by $w^{m}$. In this study, $w^{m}$ is fixed to 1. Then, if the moved position (B in Figure 15a) is not a point on the bottom boundary, it moves further up (or down) by $h^{m}$. Finally, after calculating the movement distance $\left(w^{m},h^{m}\right)$, the circular coordinate array is updated, as shown in Figure 15b, and the process of searching for the reference point commences again.

The pseudocode for calculating the difference between the current position’s coordinates and the coordinates to be moved to on the lower-right corner based on the circular coordinate array is shown in Figure 16. The difference coordinate (dx, dy) and the coordinate array $carray_{\left[i,\;len\right]}$ are used as the input parameters. The difference coordinate (dx, dy) is a parameter for returning the difference between the current position and the next position moved to. By moving a pixel (right) along the x-axis, the value of dx is set to unity (see Line 1.1). For the y-axis, because the slope fluctuates as indicated by $h^{m}$ in Figure 16a, the range of fluctuation in the y-axis is defined from +3 (maximum) to -3 (minimum) (see Lines 1.1 and 1.2) such that it can approach 0° (horizontal) to a maximum of +70° and a minimum of –70°. Thus, the candidate ranges of the difference coordinate (dx, dy) become (1,−3),(1,−2),…,(1,0), (1,1),…, and (1,3), and the position (coordinate) for moving is determined by receiving the candidate range from the check_Bottom_Side function. To determine the next position, all the pixel values corresponding to the circular coordinate array must be the same (see Lines 2.3 and 2.4), and at least one of the pixel values located at the bottom of the circular coordinate array must have a different value from the pixel value of the circular coordinate array (see Lines 2.8 and 2.9).

The pseudocode for updating the circular coordinate array is shown in Figure 17. With the difference coordinate (dx, dy) and the coordinate array carray [i, len] as the inputs, it is updated as indicated in Lines 3 and 4.

## 4. Experiment and Results

To verify the performance of the proposed method, an experiment was conducted to examine two factors: the processing speed and the accuracy. For the experiment, 78 images of an actual field (see Figure 18a, alignment-corrected images) were used, and preprocessing was performed. The images used for the experiment included images of the holes (speaker, home button, etc.) in the glass on the front of smartphones. As preprocessing, flood filling (Figure 18c) and image rotation (Figure 18d) were applied [6], after iterative binarization (Figure 18b) of the grayscale image (Figure 18a). For an objective performance analysis, the performance of the proposed method was compared with those of the line-based [7,14] and corner-based reference point extraction methods [8,9,14], among the existing reference point search algorithms [8,9,14].

### 4.1. Experimental Results for Processing Speed

Rotated images (Figure 18d) were used in the experiment, as described above. For the line-based reference point extraction method, images to which the Sobel operation had been applied were used. For the performance comparison, the end coordinate of the circle algorithm (ECCA) method adopting the midpoint circle algorithm (MCA) [12,13,15], which is another alternative to the circular-based reference point extraction method, was used. For convenience, the method proposed in Section 3 is referred to as the midpoint circular array generation method (MCAGM). Both methods are conceptually the same but differ in their concrete implementation.

The ECCA method has a similar structure to that of the MCA, in general, as shown in Figure 19. However, in the MCA, the function of drawing a circle is performed, and in the ECCA, the function of determining the reference point of an object is performed. Moreover, if the ECCA is applied, a procedure for determining whether or not to search for a reference point using the ECCA whenever a coordinate movement for searching occurs is added; this reduces the processing speed compared with that of the proposed method using a circular array (MCAGM). In the MCA, the entire circle is drawn through Lines 1.3, 1.8, and 2.1–2.4, as depicted in Figure 19a, and the coordinates of 1/8 are calculated through Lines 1.1, 1.2, and 1.4–1.7. If we applying this in the ECCA method, the reference line is determined through Lines 1.4, 1.9, 1.10, 1.11, and 2.1–2.5, as depicted in Figure 19b. The movement in the ECCA has the same structure as that of the move_Bottom_Right_Position function in Figure 16, whereas the method used to search for the bottom boundary (the boundary between the object and background) of the next position (coordinate) for moving is similar to that in Figure 19b. The code in Figure 19b determines whether the reference point is to be searched, but a method for determining the bottom boundary is added when searching for the next location (coordinate) to move. In this study, a detailed description of the movement in the ECCA is not included.

Figure 20 shows the results for the processing speed. The average processing speed was 7.81 ms for the proposed method (MCAGM), 67.91 ms for the line-based method, 496.30 ms for the Harris Corner-based method, and 1520.96 ms for the Moravec Corner-based reference point extraction method. The minimum and maximum processing speeds were 6 ms and 13 ms, respectively, for the proposed method (MCAGM); 51 ms and 271 ms, respectively, for the line-based method; 361 ms and 940 ms, respectively, for the Harris Corner-based method; and 1148 ms and 2214 ms, respectively, for the Moravec Corner-based method (see Table 1). Thus, the proposed method (MCAGM) had better performance than the line-based reference point extraction method (by 60.10 ms), the Harris Corner-based method (by 488.49 ms), and the Moravec Corner-based method (by 1512.88 ms). In addition, the overall performance of the MCAGM was better than that of the ECCA; in terms of the maximum processing time, it was better by 46 ms.

### 4.2. Experimental Results for the Accuracy

The experimental results for the accuracy reveal that the Harris Corner-based and Moravec Corner-based reference point extraction methods can accurately search for the reference point when the corner is clear, as shown in A in Figure 21a. Due to the characteristics of the test subjects (with a round corner as in B), it is difficult to clearly search for the reference point. Therefore, in the accuracy test, the performance of the proposed method (MCAGM) was compared only with that of the line-based reference point extraction method. Although accuracy verification should be performed with respect to the movement of the object and the slope of the object, in this study, only the slope of the object was measured to conduct an accuracy test. In the case of the proposed method, the slope was calculated after searching for the left-hand and right-hand reference points at the top or bottom, as shown in Figure 21b.

The angle measurement results of the proposed method (MCAGM) and the method using the line are close to the angle before rotation, as shown in Figure 22; the measurement error with respect to the actual angle is depicted in Figure 23. Overall, in the proposed method, the average error was 0.0047°, the minimum error was 0.3972°, and the maximum error was 0.1077° from the bottom; the average error was 0.0020°, the minimum error was 0.2476°, and the maximum error was 0.0792° from the top; the average error was 0.0024°, the minimum error was 0.3156°, and the maximum error was 0.0894° based on the mean ((top+bottom)/2). For the line-based reference point extraction method, the average error was 0.0016°, the minimum error was 0.0153°, and the maximum error was 0.0074° (see Table 2).

For analyzing the cause of the large error in the proposed method in detail, the images with the largest error (Image No. 46 in Figure 22) and the least error (Image No. 6 in Figure 22) were used. Comparison and analysis were performed by superimposing the same rotational line on the image to which rotation transformation was applied (e.g., image rotation 1° → line rotation 1°). Of course, the line of the object and the rotational line must coincide. In the case of the image with the smallest error (Figure 24a), there was almost no space between the object’s line and the rotational line (see A and B in Figure 24a), which means that the two lines were almost identical. However, for the image with the largest error, the space between the object and the line (see A and B in Figure 24b) was larger that than in Figure 24a, which means that the two lines did not coincide. Therefore, Image 46 was reverse rotated by the angle measured by each algorithm (see Figure 24c,d). From these results, it can be seen that the line-based method has fewer errors, as shown in Table 2, but the results of the proposed method are better according to the detailed analysis.

Overall, the experimental results of the processing speed established that the performance of the proposed method is good. In addition, in the accuracy test, the proposed methods (MCAGM and ECCA) showed better performance than the existing ones. In particular, among the proposed methods, the MCAGM showed better performance than the ECCA in terms of the processing speed.

## 5. Conclusions

With the advances in technology, the technological gap between product companies is gradually narrowing. Therefore, product design can be considered a visible strategy for differentiation. However, it is difficult to apply automated production and defect detection processes to products or parts with various designs. Therefore, a circular-based reference point extraction method for round-shaped parts was proposed in this study. To demonstrate the effectiveness of the proposed method, its processing speed was analyzed by using the ECCA, which is another implementation alternative, as the baseline. The existing methods, including the line-based, Harris Corner-based, and the Moravec Corner-based reference point extraction methods, were compared with the proposed method and analyzed in terms of the processing speed and accuracy.

The proposed method (MCAGM) showed better performance than the line-based reference point extraction method (by 60.10 ms), the Harris Corner-based method (by 488.49 ms), and the Moravec Corner-based method (by 1512.88 ms). Moreover, the overall performance of the MCAGM was better than that of the ECCA. In terms of the maximum processing time, it was better by 46 ms, in particular. The accuracy analysis results indicated that although the performance of the line-based reference point extraction method was the best among the existing methods, the performance of the proposed method (MCAGM) was better. Thus, the method proposed in this study showed better overall performance than the existing reference point extraction methods, as well as the ECCA method, which was the baseline for the proposed algorithm.

## Figures and Tables

**Figure 1 sensors-22-05859-f001:**
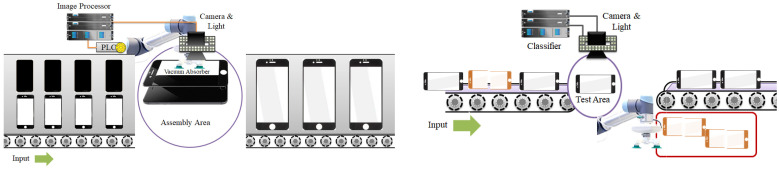
Product assembly process (**left**) and inspection process (**right**).

**Figure 2 sensors-22-05859-f002:**
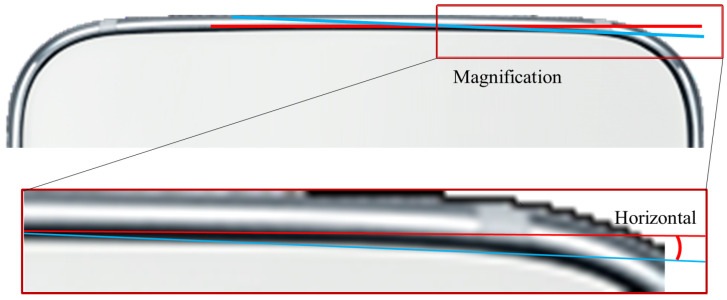
Example of a round design.

**Figure 3 sensors-22-05859-f003:**
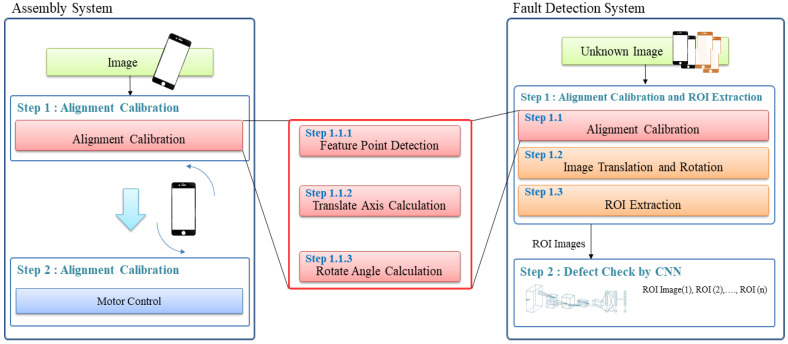
Detailed structure of the product assembly process and defect detection system.

**Figure 4 sensors-22-05859-f004:**
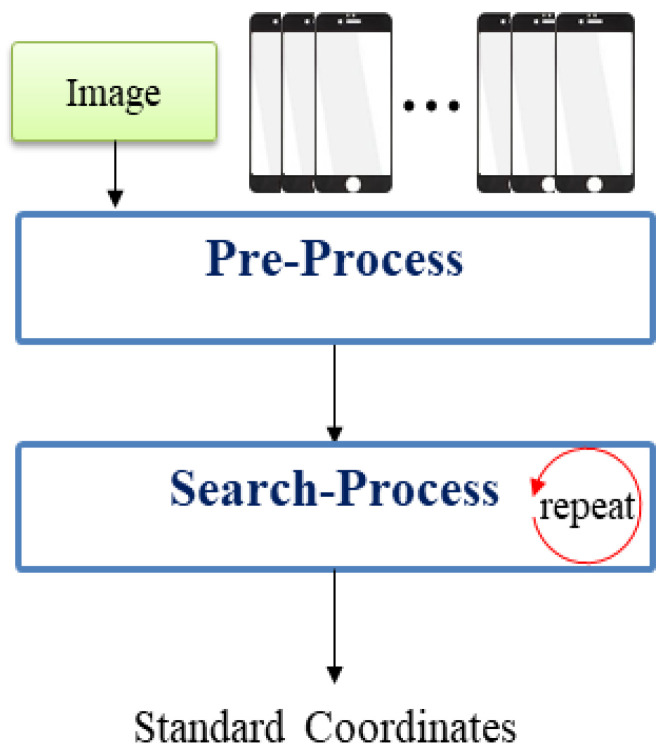
Process of extracting the reference point of the product.

**Figure 5 sensors-22-05859-f005:**
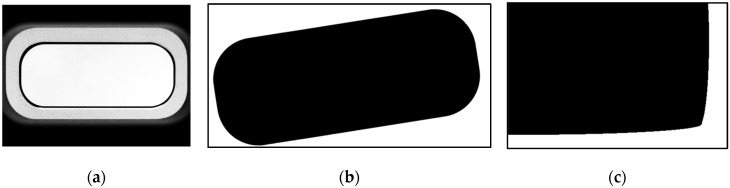
Targets for alignment correction: (**a**) smartphone hole; (**b**) object with curved and linear components; (**c**) object with curved components.

**Figure 6 sensors-22-05859-f006:**
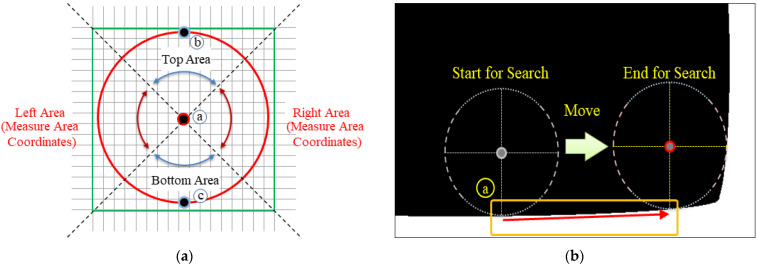
Concept of the circle-based reference point search algorithm. (**a**) Circle and function for reference point extraction (ⓐ: center point A, ⓑ: top point B, ⓒ: bottom point C); (**b**) Object with curved components (ⓐ: moving path A).

**Figure 7 sensors-22-05859-f007:**
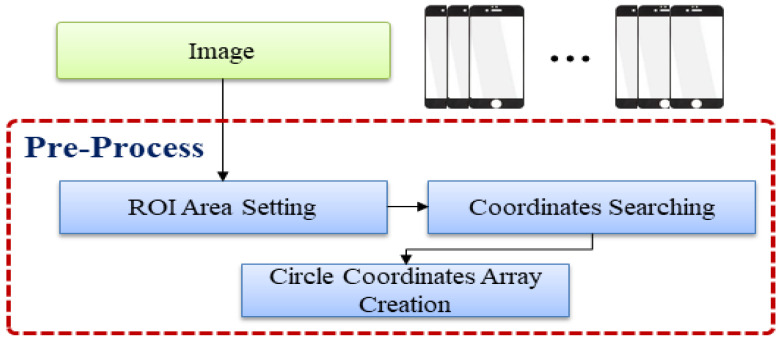
Preprocessing.

**Figure 8 sensors-22-05859-f008:**
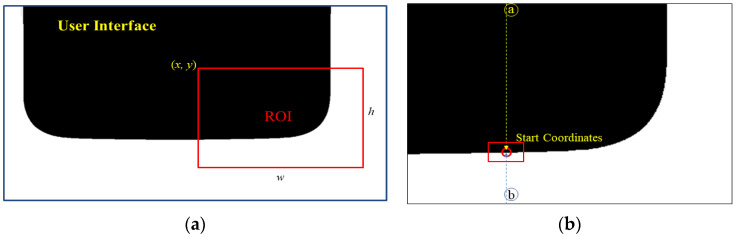
Preprocessing steps. (**a**) Setting the ROI. (**b**) Starting point search (ⓐ: search position of point A, ⓑ: search position of point B).

**Figure 9 sensors-22-05859-f009:**
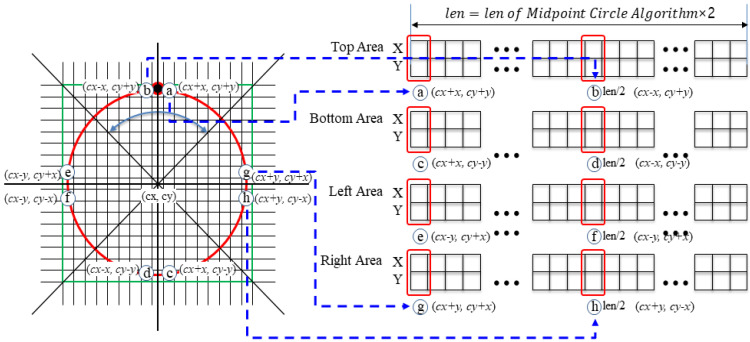
Generation of a circular coordinate array.

**Figure 10 sensors-22-05859-f010:**
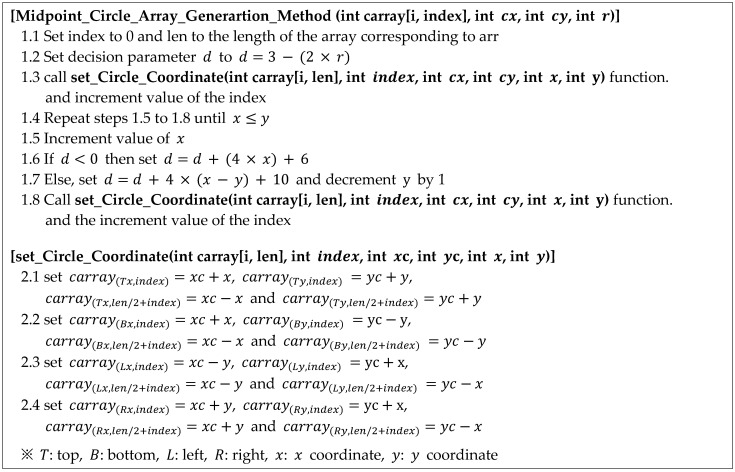
Proposed algorithm (circular coordinate array generation).

**Figure 11 sensors-22-05859-f011:**
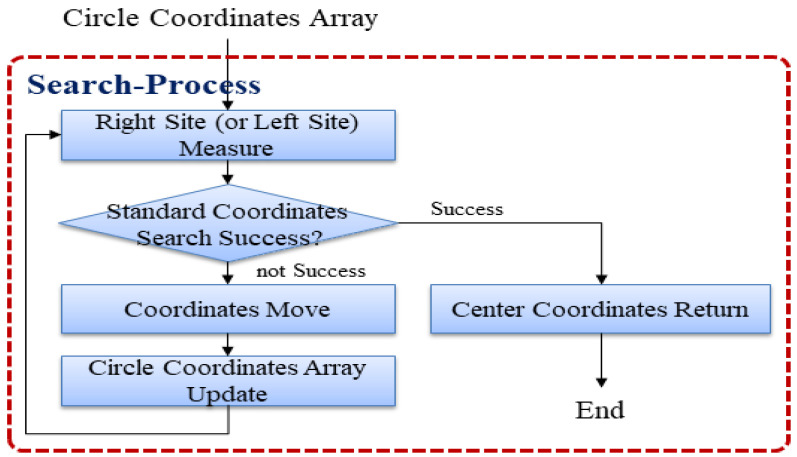
Starting point search process.

**Figure 12 sensors-22-05859-f012:**
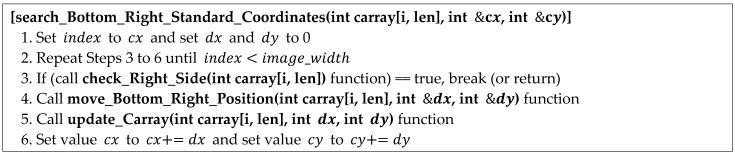
Proposed algorithm: reference point search for the lower-right corner.

**Figure 13 sensors-22-05859-f013:**
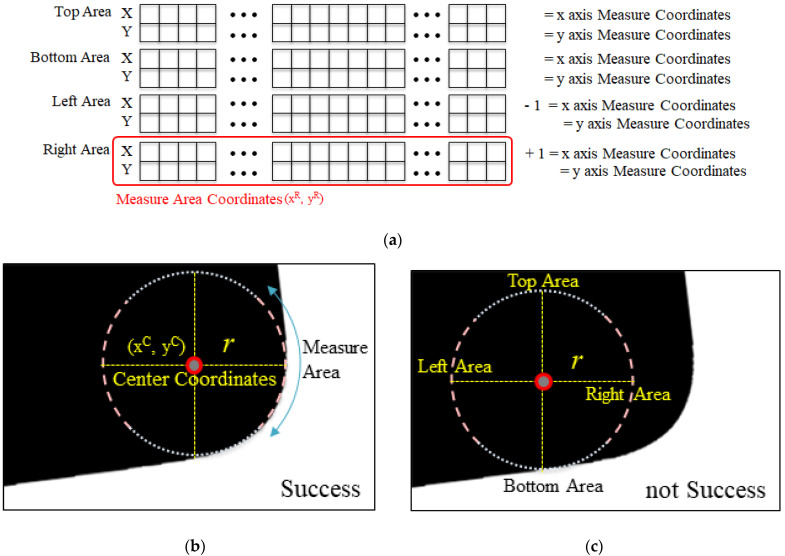
Reference point extraction method (lower-right area). (**a**) Right-hand search area of the circular coordinate array. (**b**) Successful search. (**c**) Search continued.

**Figure 14 sensors-22-05859-f014:**
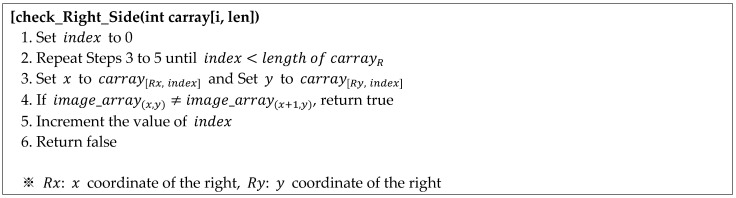
Proposed algorithm (reference point check for the lower-right corner).

**Figure 15 sensors-22-05859-f015:**
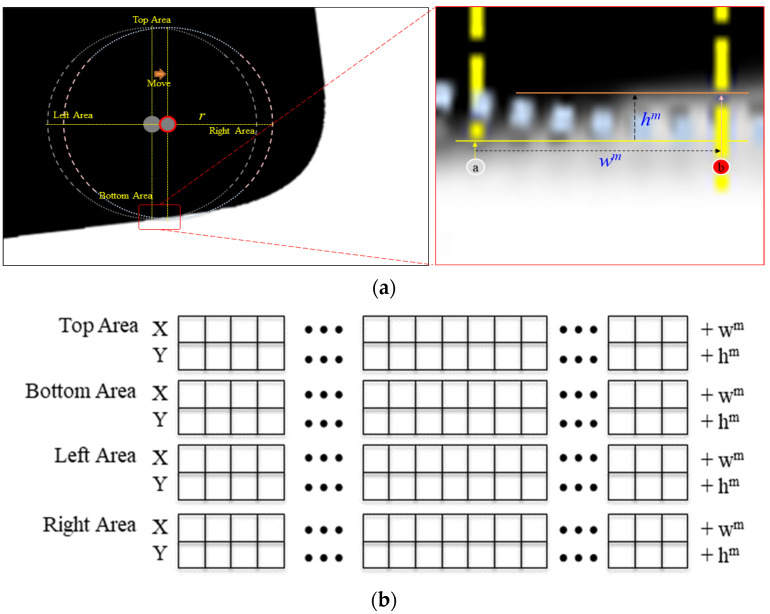
Movement of the coordinates and updating of the circular coordinate array. (**a**) Example of coordinate movement (ⓐ: the current position A, ⓑ: the moved position B). (**b**) Updating the circular coordinate array.

**Figure 16 sensors-22-05859-f016:**
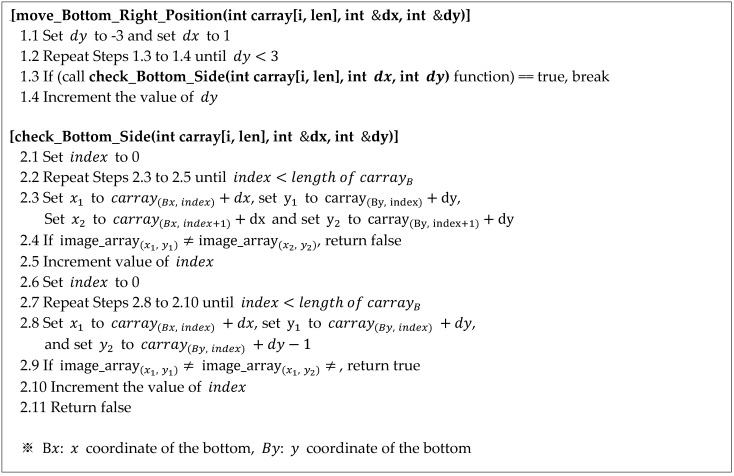
Proposed algorithm for movement to the right and bottom.

**Figure 17 sensors-22-05859-f017:**
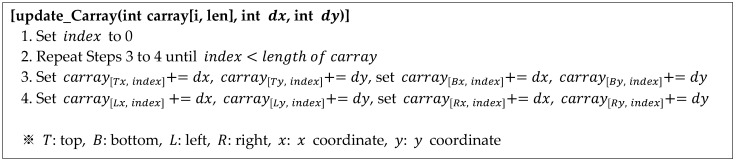
Proposed algorithm for updating the circular coordinate array.

**Figure 18 sensors-22-05859-f018:**
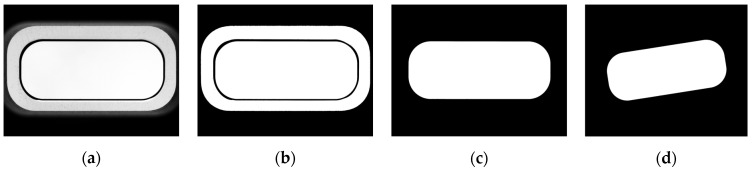
Experimental image sample and preprocessed image for the experiment: (**a**) experimental image sample; (**b**) iterative binarization; (**c**) flood filling; (**d**) rotation.

**Figure 19 sensors-22-05859-f019:**
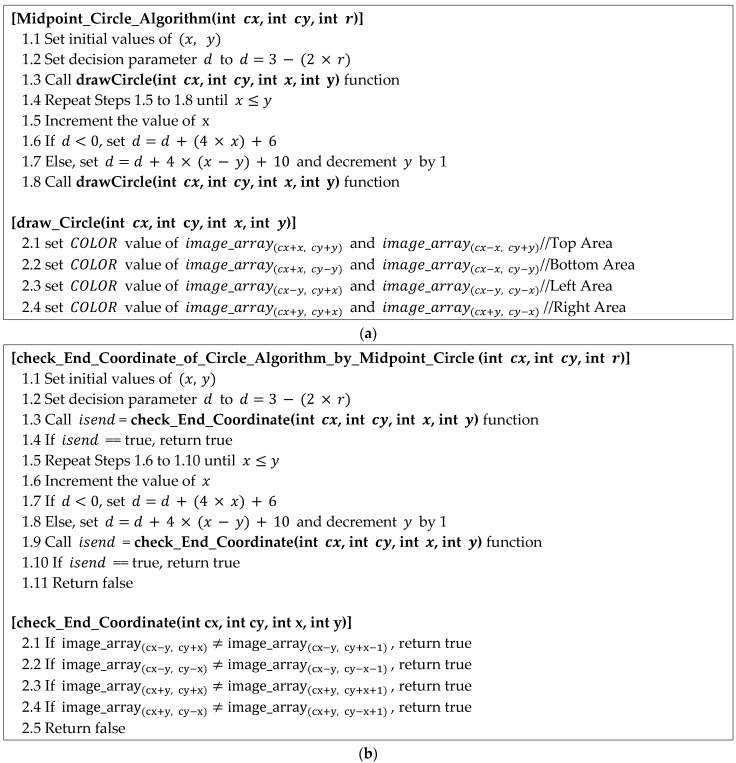
Midpoint circle algorithm vs. the check End Coordinate of the Circle algorithm. (**a**) Midpoint circle algorithm. (**b**) Check End Coordinate of the Circle algorithm for the midpoint circle.

**Figure 20 sensors-22-05859-f020:**
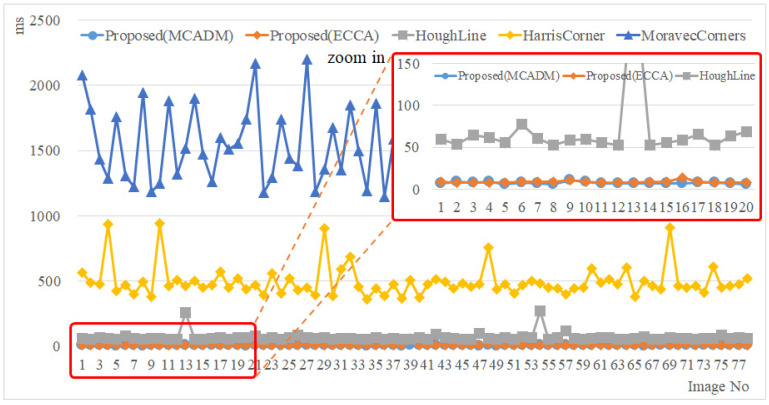
Results for processing speed (release mode).

**Figure 21 sensors-22-05859-f021:**
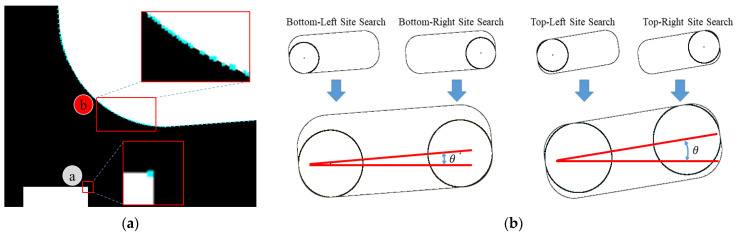
Experimental image sample and preprocessed image for the experiment. (**a**) Results of the Harris and Moravec Corner methods (ⓐ: corner A, ⓑ: round corner B). (**b**) Slope calculation in the proposed method).

**Figure 22 sensors-22-05859-f022:**
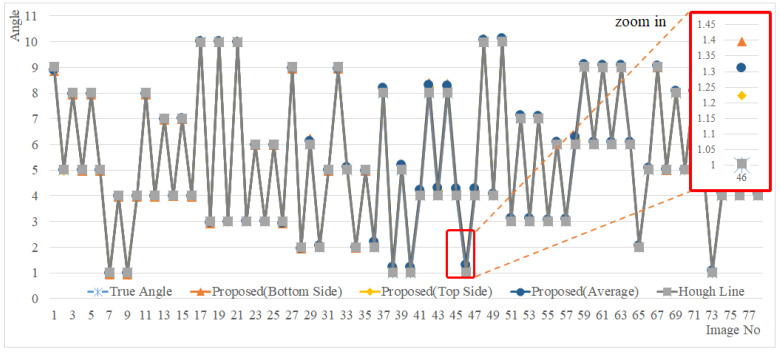
Angle measurement results.

**Figure 23 sensors-22-05859-f023:**
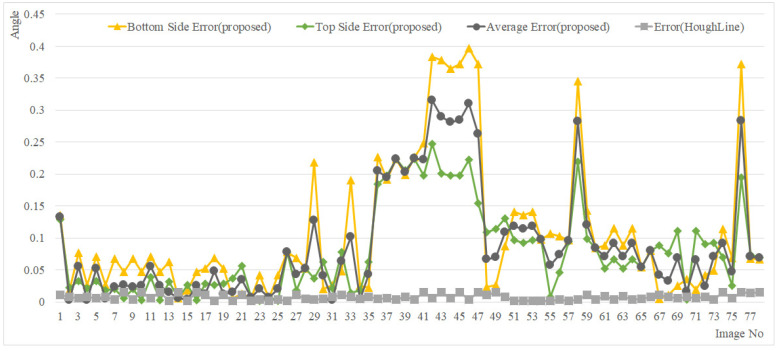
Measurement error.

**Figure 24 sensors-22-05859-f024:**
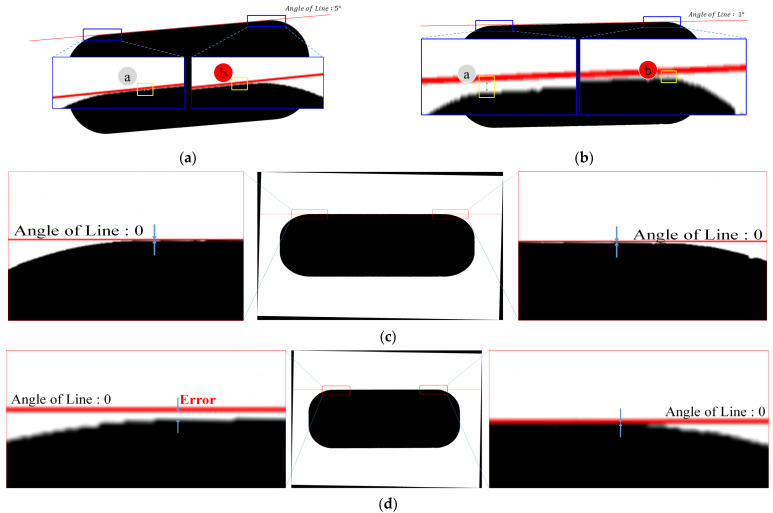
Analysis results: (**a**) Image No. 6 (ⓐ: error A, ⓑ: error B); (**b**) Image No. 46 (ⓐ: error A, ⓑ: error B). (**c**) After reverse rotation by the angle measured by the proposed method. (**d**) After reverse rotation by the angle measured by the line-based method.

**Table 1 sensors-22-05859-t001:** Comparison of the processing times.

Unit: ms	Proposed 1 (MCAGM)	Proposed 2 (ECCA)	Hough Line	Harris Corner	Moravec Corner
Average	**7.81**	9.82	67.91	496.30	1520.69
Min	**6**	7	51	361	1148
Max	**13**	59	271	940	2214

**Table 2 sensors-22-05859-t002:** Slope error comparison.

Unit: Angle (°)	Bottom-Side Error (Proposed)	Top-Side Error (Proposed)	Average Error (Proposed)	Error (Hough Line)
Average	0.0047	0.0020	0.0024	0.0016
Min	0.3972	0.2476	0.3156	0.0153
Max	0.1077	0.0792	0.0894	0.0074

## Data Availability

The data presented in this study are available on request from the corresponding author.

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
