# Peer review of "A Circular-Based Reference Point Extraction Method for Correcting the Alignment of Round Parts"

_sensors, 2022, doi:10.3390/s22155859_

Round 1

Reviewer 1 Report

This article is interesting but requires some changes.

1. Please specify in numbers (eg in %) how the given method is more effective. 2. Is the fillet radius related to the method performance?

Reviewer 2 Report

It is an ultra-specialized paper whose interest for the widest audience of readers may be low. In any case, the method developed and the results obtained certainly worth publishing. The text describes in too much detail all the steps of the reference point finding in a alignment procedure. Authors should make an effort to reduce the size of the text, probably by creating an appendix in which all passages are described in detail. The list of individual steps would remain in the text with a reference to their detailed description.

The results certainly appear good showing how the proposed algorithm strongly reduces the calculation times while maintaining a high accuracy. The point that must certainly be corrected is the excessive number of self-citations by the authors (4 out of 15 total entries).

The abstract is fine as well as the figures. I believe that the paper can be published after a good revision of the text.
